# Clinical Characteristics and Epidemiological Features of Hepatitis E Virus Infection Among People Living with HIV in Shanghai, China

**DOI:** 10.3390/v17081038

**Published:** 2025-07-25

**Authors:** Conglin Zhao, Yuanyuan Ji, Shuai Tao, Mengxin Lu, Yi Zhang, Weixia Li, Shuangshuang Sun, Han Zhao, Weijia Lin, Yuxian Huang, Qiang Li, Chong Chen, Liang Chen

**Affiliations:** 1Department of Liver Disease, Shanghai Public Health Clinical Center, Fudan University, Shanghai 200437, China; lin15993001733@163.com (C.Z.); jiyuanyuan@shaphc.org (Y.J.); lumxxin@foxmail.com (M.L.); zhangyi117@shaphc.org (Y.Z.); yxhuang@fudan.edu.cn (Y.H.); liqiang66601@163.com (Q.L.); 2Scientific Research Center, Shanghai Public Health Clinical Center, Fudan University, Shanghai 200437, China; taoshuai@shaphc.org; 3Department of Infectious Diseases, Shanghai Public Health Clinical Center, Fudan University, Shanghai 200437, China; liweixia@shaphc.org; 4Department of Liver Disease Center, Shanghai Public Health Clinical Center, Fudan University, Shanghai 200437, China; 13818680249@163.com; 5Department of Liver Intensive Care Unit, Shanghai Public Health Clinical Center, Fudan University, Shanghai 200437, China; zhaohan@shaphc.org; 6Department of Hepatobiliary Medicine, Shanghai Public Health Clinical Center, Fudan University, Shanghai 200437, China; linweijia@shaphc.org

**Keywords:** hepatitis E virus, HIV, seroprevalence, risk factors

## Abstract

Hepatitis E virus (HEV) poses a significant public health concern, particularly among immunocompromised populations. This study aimed to investigate HEV seroprevalence, clinical characteristics, and associated risk factors in people living with HIV (PLWH) in Shanghai, China. A retrospective analysis was conducted on serum IgG and IgM antibodies specific to HEV in 670 PLWH and 464 HIV-negative health-check attendees. The overall anti-HEV seropositivity rate among PLWH was 30.15% (202/670, 95% CI 26.68–33.62), with an IgG positivity rate of 30.00% (201/670, 95% CI 26.53–33.47). IgM positivity was observed in 1.19% (8/670, 95% CI 0.59–2.39) of PLWH, and dual IgM/IgG positivity was observed in 1.04% (7/670, 95% CI 0.50–2.16) of PLWH. The seropositivity rate of anti-HEV IgG in the HIV-negative health-check attendees was 17.67% (82/464, 95% confidence interval: 14.20–21.14), with no IgM positivity, which was significantly lower than that in PLWH (χ^2^ = 22.84, *p* < 0.001). Univariate and multivariate analyses identified advanced World Health Organization (WHO) HIV stage (III/IV) as an independent risk factor for HEV co-infection (*p* < 0.05). Notably, no significant associations were observed with age, gender, CD4 count, or liver function parameters. These findings underscore the importance of implementing HEV screening protocols and developing targeted preventive strategies for PLWH.

## 1. Introduction

Hepatitis E virus (HEV), a single-stranded, non-enveloped RNA virus that is transmitted primarily via the fecal-oral route, is a major etiological agent of acute viral hepatitis globally [1,2]. Approximately 20 million HEV infections occur annually worldwide, resulting in around 70,000 fatalities [3]. Although immunocompetent hosts typically develop self-limiting hepatitis, immunocompromised patients—including organ transplant recipients [4], patients with hematological malignancies [5,6], and people living with HIV (PLWH) [7,8]—are at risk for chronic infection and cirrhosis.

At the same time, HIV remains a global public health challenge, with annual mortality still exceeding several hundred thousand [9]. The progressive immune depletion in PLWH increases vulnerability to opportunistic infections and atypical disease progression [10], potentially altering HEV persistence and clinical manifestations [11]. The geographical overlap between HIV-endemic regions and HEV-hyperendemic zones raises significant concerns about co-infection epidemiology. However, research on HIV and hepatitis E virus co-infection is currently limited. Although China is endemic for HEV, data on the serological prevalence patterns and associated risk factors among this vulnerable population in Shanghai remain scarce.

This study aims to (1) determine the seroprevalence of HEV among PLWH in Shanghai; (2) compare seroprevalence rates with HIV-negative health-check attendees; (3) characterize clinical profiles of co-infected individuals; and (4) identify epidemiological determinants of HEV infection. Through this comprehensive analysis, we seek to elucidate the complex interactions between HIV and HEV infections in Shanghai, informing targeted surveillance strategies and clinical management protocols for this vulnerable population.

## 2. Materials and Methods

### 2.1. Study Design and Population

Between 2018 and 2024, our clinic followed 8468 adult PLWH (≥18 years); of these, 714 (8.4%) underwent anti-HEV testing as part of routine annual liver function surveillance, independent of clinical hepatitis symptoms. A total of 44 patients were excluded due to incomplete clinical data. Ultimately, 670 PLWH were included in this study (Figure 1). Controls comprised 464 HIV-seronegative adults who attended the hospital’s annual health check program. All of these individuals voluntarily requested anti-HEV antibody testing as part of their health check due to increased awareness of hepatitis prevention. The control group was age- and sex-matched to the PLWH group, and all participants reported no chronic liver disease or immunodeficiency.

### 2.2. Data Collection

Demographic and clinical parameters were extracted from electronic medical records. These included epidemiological characteristics such as age, gender, ethnicity, and marital status. Hepatic profiles were also collected, comprising albumin (ALB), alanine aminotransferase (ALT), aspartate aminotransferase (AST), and total bilirubin (TBIL). HIV-specific markers were documented, including CD4 count, antiretroviral therapy (ART) regimen, and WHO clinical stage. Additionally, comorbidity data were gathered, focusing on hepatitis B virus (HBV)/hepatitis C virus (HCV) co-infection status and liver cirrhosis.

### 2.3. Serological Assays

Serum anti-HEV IgM and IgG antibodies were detected using commercial enzyme-linked immunosorbent assay (ELISA) kits (Wantai Biological Pharmacy, Beijing, China) according to the manufacturer’s specifications. The assay demonstrated specific performance characteristics for each antibody type: Anti-HEV IgG showed high sensitivity at 97.7% and excellent specificity at 99.6%, while Anti-HEV IgM exhibited strong sensitivity at 93.2% and robust specificity at 97.8% [12]. Liver function parameters were analyzed on Hitachi 7600 automated analyzers (Hitachi High-Technologies Corporation, Tokyo, Japan) using standard enzymatic methods. The CD4 count was quantified in peripheral blood specimens from PLWH using flow cytometric analysis.

HEV RNA was detected using the reverse transcription-quantitative polymerase chain reaction (RT-qPCR) technique for samples positive for anti-HEV IgM antibodies. RNA extraction and RT-qPCR procedures were essentially the same as described previously, with a lower limit of detection of 500 copies/mL [13,14]. For molecular detection, HEV RNA was extracted from the serum using a QIAamp Viral RNA Mini Kit (Qiagen, Hilden, Germany). Quantitative RT-PCR was performed on Applied Biosystems 7500 platforms using Vazyme One-Step RT-PCR kits (Vazyme, Nanjing, China). Primer and probe sequences are detailed in Appendix A.

### 2.4. Statistical Analysis

The analyses were conducted using SPSS 20.0 and R 4.3.2. Continuous variables were expressed as mean ± standard deviation (SD) (for normally distributed data) or median [interquartile range (IQR)] (for non-normally distributed data) and were assessed by Student’s *t*-test or Mann–Whitney U-test, respectively. Categorical variables are presented as counts and percentages and compared using a chi-square test or Fisher’s exact test. Annual HEV seroprevalence trends (2018–2024) were examined using the Cochran–Armitage trend test. To mitigate detection bias from testing fluctuations during the COVID-19 pandemic, inverse-variance weighted linear regression was performed (weights = 1/variance of annual estimates). Age-specific patterns were analyzed using polynomial regression with quadratic term testing for non-linearity. Following significant non-linearity, Bonferroni-adjusted pairwise comparisons between age groups were conducted. Joinpoint regression identified significant inflection points in age trends. Associations between anti-HEV antibody positivity and potential risk factors were evaluated in univariate and multivariate logistic regression models. Odds ratio (OR) and 95% confidence interval (95% CI) were calculated. We calculated variance inflation factors (VIFs) for all variables in our multivariate model. Results from two-sided tests with *p* < 0.05 were considered statistically significant.

## 3. Results

### 3.1. Demographic and Clinical Characteristics

The demographic, laboratory, and clinical characteristics of PLWH are shown in Table 1. The study cohort comprised 670 PLWH with a median age of 40 years (IQR 32–53), predominantly male (87.76%, 588/670) and of Han ethnicity (98.96%, 663/670). Clinical profiling revealed a median CD4 count of 211.9 cells/μL (IQR 45.2–473.7), with 61.34% (411/670) classified as WHO stage I/II. Viral hepatitis co-infections were observed in 6.87% (46/670; HBV) and 4.03% (27/670; HCV) of participants. Hepatic parameters demonstrated median levels of albumin (38.3 g/L, IQR 31.8–44.5), ALT (30.0 U/L, IQR 16.0–79.4), AST (30.0 U/L, IQR 20.0–63.8), and total bilirubin (10.4 μmol/L, IQR 7.0–16.7). Comparative analysis identified significant disparities between HEV-seropositive and seronegative groups, with the former exhibiting an older median age (47 vs. 37 years, *p* < 0.001) and higher total bilirubin levels (11.6 vs. 9.85 μmol/L, *p* = 0.003).

### 3.2. Anti-HEV Seroprevalence

The overall anti-HEV seropositivity rate among PLWH was 30.15% (202/670, 95% CI 26.68–33.62), with an IgG positivity rate of 30.00% (201/670, 95% CI 26.53–33.47). IgM positivity was observed in 1.19% of PLWH (8/670, 95% CI 0.59–2.39), and dual IgM/IgG positivity was observed in 1.04% of PLWH (7/670, 95% CI 0.50–2.16) (Table 2). Among IgM-positive cases (*n* = 8), 75.00% (6/8) presented with symptoms of acute hepatitis (jaundice and dark urine) and elevated liver transaminases, suggesting they might be in an acute or recent state of infection. HEV RNA was not detected in patients who were positive for anti-HEV IgM in this study. Detailed clinical data are presented in Table 3.

Among 464 HIV-negative health-check attendees, 82 individuals were positive for anti-HEV antibodies (all were positive for anti-HEV IgG), with a positivity rate of 17.67% (82/464, 95% CI 14.20–21.14%). No one in the HIV-negative health-check attendees was positive for anti-HEV IgM. There was a significant difference in HEV seropositivity rates between HIV-infected individuals and the HIV-negative health-check attendees (χ^2^ = 22.84, *p* < 0.001).

### 3.3. HEV Seroprevalence Patterns

Comprehensive analysis revealed stable HEV seroepidemiological patterns, without significant temporal, age-related, or gender-based variations (Figure 2). Annual HEV seroprevalence rates demonstrated stability from 2018 to 2024 (Cochran–Armitage trend test, *p* = 0.15), although the testing frequency exhibited notable fluctuations. The observed reduction in testing volume during 2020–2022 (71 tests in 2020, 64 in 2021, and 33 in 2022) coincided with the COVID-19 pandemic period, during which routine clinical surveillance activities were substantially disrupted. Despite this pandemic-related testing reduction, prevalence estimates remained consistent (2020: 33.8%, 95% CI: 23.0–46.0; 2021: 21.9%, 95% CI: 12.5–34.0; 2022: 27.3%, 95% CI: 13.3–45.5). To address potential detection bias arising from variable testing intensity, particularly during the pandemic years, we conducted inverse-variance weighted regression analysis. This sensitivity analysis confirmed trend stability (β = −0.008, 95% CI: −0.022–0.006, *p* = 0.18), supporting the absence of significant temporal variation independent of testing fluctuations.

Regarding age-specific patterns, seroprevalence distribution showed a non-monotonic configuration with peak rates in 40–49 year-olds (35.0%, 95% CI: 27.1–43.6) and those ≥60 years (31.4%, 95% CI: 22.7–41.2). Statistical validation through polynomial regression identified a significant quadratic component (β = 0.021, *p* = 0.04), confirming deviation from linearity. Post-hoc pairwise comparisons with Bonferroni adjustment revealed significantly higher seroprevalence in 40–49 year-olds compared to 30–39 year-olds (Δ = 6.0%, OR = 1.38, 95% CI: 1.02–1.87, *p* = 0.04). Joinpoint regression further identified significant inflection points at 45 years (95% CI: 42–48, *p* = 0.03) and 65 years (95% CI: 61–69, *p* = 0.05), statistically corroborating the observed bimodal pattern. Gender differences were non-significant both overall and within all age strata (all *p* > 0.05).

CD4 count was categorized into three levels: <200 cells/μL, ≥200 to <400 cells/μL, and ≥400 cells/μL. Stratification by CD4 count was performed to assess the relationship between CD4 count and HEV seropositivity rates. There was no significant difference in HEV seropositivity rates among the groups (CD4 count < 200 cells/μL: 88/323 [27.24%]; 200 ≤ CD4 count < 400 cells/μL: 47/139 [33.81%]; CD4 count ≥ 400 cells/μL: 67/208 [32.21%], *p* = 0.26). To further explore the potential association between CD4 count and HEV infection risk, we plotted the relationship between CD4 count and the estimated probability of HEV infection (Figure 3). The graph aimed to visualize whether changes in CD4 count influence HEV infection risk, particularly in immunocompromised individuals. A slight upward trend was observed as CD4 count increased, although this trend was not statistically significant.

### 3.4. Risk Factor Analysis

We systematically evaluated risk factors for HEV co-infection in PLWH through univariate and multivariate logistic regression analyses (Table 4). Univariate analysis revealed a significant association between WHO HIV stage III/IV and anti-HEV antibody positivity (OR 1.422, 95% CI 1.016–1.988, *p* = 0.040). Anti-HEV positivity, age, gender, ethnicity, marital status, CD4 count, ART regimen, and co-infection were not associated with other hepatitis viruses or levels of ALB, ALT, AST, and total bilirubin (*p* > 0.05). Notably, these associations could not be determined (OR = 0.000) in non-Han ethnic groups due to a limited sample size, requiring cautious interpretation.

In the multivariate analysis conducted for all patients, WHO staging again emerged as an independent variable associated with anti-HEV antibody positivity (Table 4). Compared to early-stage patients, advanced WHO HIV stage (III/IV) was associated with a higher likelihood of prior HEV exposure (OR 1.566, 95% CI 1.068–2.297, *p* = 0.022). The VIFs for WHO stages and CD4 count were 1.3 and 1.6, respectively, indicating acceptable levels of collinearity (VIF < 5). No other variables in this model showed significant associations with anti-HEV antibody status (*p* > 0.05).

## 4. Discussion

This study provides key novel insights into HEV epidemiology among PLWH in Shanghai: (1) We report an anti-HEV IgG seroprevalence rate of 30.00%. (2) Despite this significant seropositivity, HEV RNA remained undetectable in all anti-HEV IgM-positive patients (1.19% of the cohort), suggesting acute/recent infection is uncommon and chronic infection, at least at detectable levels, is rare in this population. (3) Using Wantai ELISA, we demonstrated a significantly higher HEV seropositivity rate in PLWH (30.15%) compared to HIV-negative health-check attendees (17.67%), indicating increased susceptibility. (4) Advanced WHO HIV clinical stage (III/IV), but not CD4 count, emerged as a significant independent risk factor for prior HEV exposure. These findings fill a critical gap in the understanding of HEV co-infection in this major Chinese urban HIV cohort.

HEV has emerged as a leading cause of acute viral hepatitis globally [15,16,17]. While immunocompetent individuals typically experience asymptomatic seroconversion with minimal risk of chronic infection [18], immunosuppressed populations demonstrate a spectrum of clinical manifestations, including acute hepatitis and persistent chronic infection or viral reactivation [4,19,20]. Epidemiological data on HIV–HEV co-infection are scarce, with reported seroprevalence rates varying considerably across studies and geographical regions. These discrepancies likely stem from substantial differences in sample sources, study populations, and diagnostic methodologies. Previous cross-sectional studies report HEV seropositivity rates among PLWH ranging from 1.0% in Scotland to 71% in Zambia [21,22,23,24,25,26] (Appendix A). Notably, HEV infection in immunosuppressed hosts can become chronic and rapidly progress to cirrhosis [27,28], and HEV seropositivity in PLWH may be underestimated due to impaired seroconversion kinetics [12]. Despite the significance of HEV in this context, research on HIV and HEV co-infection remains limited in Shanghai, China.

The observed IgG seroprevalence (30.0%) exceeds rates reported in European and North American populations but aligns with ranges documented across Africa and Asia [7,8,11,12,21,22,23,24,25,26,27,29,30,31,32,33,34,35,36,37,38,39,40,41,42,43,44,45,46,47,48,49,50,51,52,53,54,55,56,57,58,59,60,61,62,63,64,65,66,67,68,69,70]. Provincial comparisons within China reveal Shanghai’s prevalence among PLWH resembles Anhui (31.3%) and exceeds Xinjiang (25.5%) but remains below Henan (44.2%), Yunnan (56.8%), and Zhejiang (41.1%) [71]. Crucially, anti-HEV antibody testing across these Chinese studies, including ours, utilized the same Wantai ELISA kits, eliminating inter-study variability due to assay differences. Consistent with these regional patterns, HEV RNA remained undetectable in all anti-HEV IgM-positive patients, mirroring reports from other Chinese cities of low IgM seropositivity (0.3–0.78%) and absent viremia [64,71]. Consequently, while HEV seropositivity is relatively common among PLWH, chronic HEV infection appears to be a rare manifestation of chronic liver disease in this population. However, the true prevalence of chronic HEV infection may be underestimated due to the limitations in detecting low-level viremia. Given the 500 copies/mL threshold, low-level viremia may have gone undetected; so, a negative RNA result does not definitively exclude chronic HEV infection in PLWH. Future studies would benefit from ultrasensitive PCR assays to detect occult viremia in immunocompromised populations.

The question of whether HIV infection heightens susceptibility to HEV remains inconclusive. Studies incorporating HIV-negative controls have yielded conflicting results regarding differences in HEV exposure rates between PLWH and HIV-negative controls [21,26]. While some suggest PLWH may be more susceptible to HEV infection [8,29], others report no significant difference compared to HIV-negative controls [21,26,41]. In this study, the HEV seropositivity rate among HIV-negative health-check attendees was 17.67%, which is very close to the HEV seropositivity rate (18.02%) observed in the general population, according to the monitoring data from Chinese health examination centers from 2017 to 2022 [72]. Critically, we observed a significantly higher HEV seropositivity rate among PLWH (30.15%) compared to HIV-negative health-check attendees (17.67%). These findings suggest an increased vulnerability of PLWH to HEV infection, necessitating heightened clinical vigilance. This raises important questions regarding whether PLWH face higher exposure risks or possess increased biological susceptibility, warranting further mechanistic investigation.

HEV seroprevalence demonstrated relative stability across temporal, demographic, and immunological strata. No significant longitudinal trends were observed between 2018 and 2024 (*p* = 0.15), and no significant age-dependent gradients (*p* = 0.53) or gender disparities were detected. Although age-stratified analysis indicated a progressive increase in seroprevalence from 25.00% (18–29 years) to 35.00% (40–49 years)—consistent with established age-dependent exposure patterns in both immunocompetent individuals and PLWH [30,52,57,73]—multivariate regression failed to identify age as an independent risk factor (*p* > 0.05), suggesting potential cohort-specific effects.

Critically, advanced WHO HIV clinical stage (III/IV) emerged as an independent risk factor for HEV seropositivity in both univariate and multivariate analyses, highlighting that advanced WHO HIV stage (III/IV) was associated with a higher likelihood of prior HEV exposure. This association persisted even after adjustment for CD4 count in multivariate models, while CD4 count itself showed no independent correlation. This key dissociation implies that susceptibility is mediated by pathophysiological features of advanced HIV disease extending beyond CD4 count. WHO staging captures cumulative immune damage (e.g., chronic mucosal barrier disruption and gut-associated lymphoid tissue (GALT) impairment), whereas CD4 count primarily reflects current immunologic status [74]. Given the predominantly enteric transmission route of HEV, the persistent intestinal compromise characteristic of late-stage HIV may constitute a salient biological risk factor, irrespective of recent CD4 recovery following antiretroviral therapy [75,76]. However, the cross-sectional design precludes inference about directionality, and unrecognized chronic HEV infection could itself contribute to HIV disease progression. These findings underscore the clinical imperative for enhanced HEV vigilance in advanced HIV disease and warrant further investigation into the specific mechanisms underlying this association.

Several limitations of this study warrant careful consideration. The cross-sectional design, while providing valuable prevalence data, inherently limits causal inference regarding risk factors for HEV infection in PLWH, underscoring the need for longitudinal studies to establish temporal relationships and potential causality. Additionally, the absence of detailed exposure history data (e.g., dietary habits, zoonotic contact, water sources, transfusion history, occupation, and household income) hinders the identification of specific HEV transmission routes in this population and may introduce potential confounding. This limitation necessitates more comprehensive data collection in future studies. Critically, our complete reliance on serological diagnosis constitutes an important constraint. Immunocompromised individuals, including PLWH, may exhibit impaired or delayed humoral immune responses to HEV infection [5,6,7,77,78], potentially leading to false-negative serological results and an underestimation of true exposure rates. Furthermore, while HEV RNA testing was performed for all IgM-positive individuals, the lack of universal RNA testing across all seropositive (IgG+) participants precludes a comprehensive assessment of chronic HEV infection prevalence in this cohort. Finally, the single-center recruitment strategy may limit the generalizability of the findings to broader populations or different geographical regions. These limitations highlight the importance of cautious interpretation and indicate key areas for improvement in future studies on HIV–HEV co-infection.

Future research directions emerging from this study are multifaceted. Prospective cohort studies are essential to elucidate the long-term outcomes of HIV–HEV co-infection. In-depth exploration of transmission routes and risk factors specific to HEV infection in PLWH is crucial for developing targeted prevention strategies. Given the availability of HEV vaccines with documented immunogenicity, evaluating their efficacy and safety in PLWH represents a promising approach to reducing co-infection rates. Additionally, future studies should incorporate HEV-specific T-cell assays or antigen detection methods to identify seronegative infections, thereby providing a more comprehensive assessment of HEV exposure in immunocompromised populations.

## 5. Conclusions

In summary, this study demonstrates that PLWH in Shanghai exhibit advanced WHO HIV stage (III/IV), which was associated with a higher likelihood of prior HEV exposure. This underscores the necessity of incorporating targeted HEV screening awareness and prevention strategies into HIV care programs, particularly for patients with advanced disease. These findings provide a foundation for future research and potential refinements in clinical practice, aiming to enhance care and prognosis for affected individuals.

## Figures and Tables

**Figure 1 viruses-17-01038-f001:**
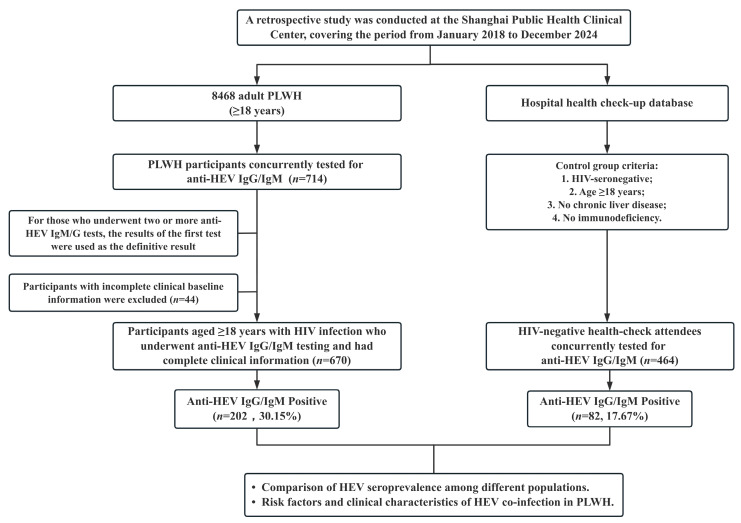
Flowchart of the study participants. HEV, Hepatitis E Virus; HIV, Human Immunodeficiency Virus; PLWH, People Living with HIV; IgM, Immunoglobulin M; IgG, Immunoglobulin G.

**Figure 2 viruses-17-01038-f002:**
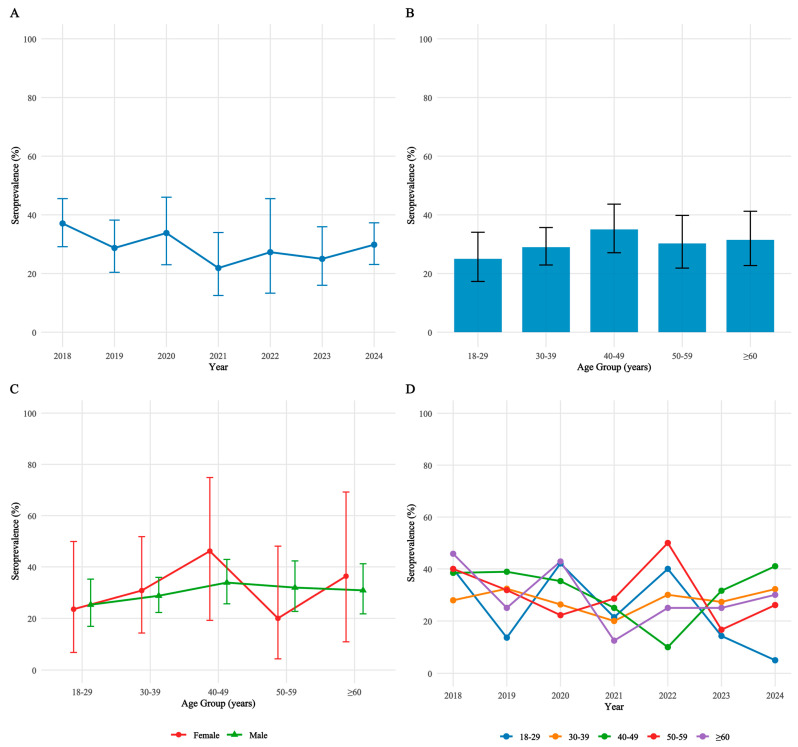
HEV seroprevalence patterns in PLWH. (**A**). Temporal trend of HEV seroprevalence from 2018 to 2024. Error bars indicate 95% confidence intervals. (**B**). Age-specific HEV seroprevalence. Error bars indicate 95% confidence intervals. (**C**). Age- and gender-stratified seroprevalence. Red: female; green: male. (**D**). Temporal trends by age cohort. HEV, Hepatitis E Virus; HIV, Human Immunodeficiency Virus; PLWH, People Living with HIV.

**Figure 3 viruses-17-01038-f003:**
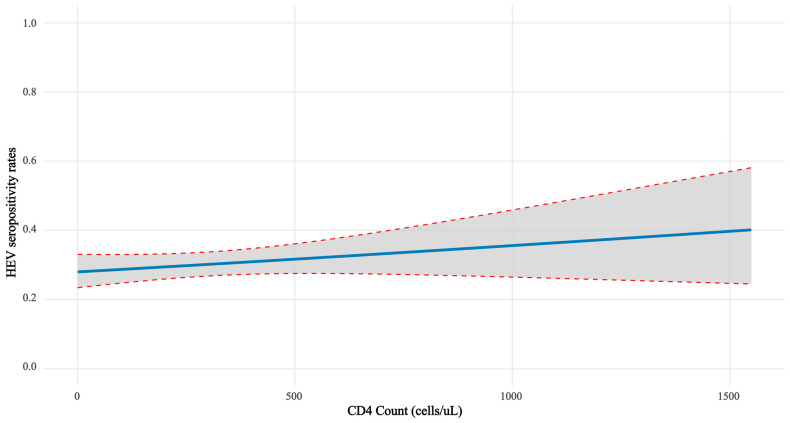
Relationship between CD4 count and estimated probability of HEV infection. The solid line represents the estimated probability of HEV infection across different CD4 counts. The dashed lines indicate the 95% confidence interval for this estimation. Model: logistic regression (glm, binomial family). Curve: predicted probability (solid line) with 95% Wald CI on logit scale back-transformed via inverse logit (shaded band). Fit: AIC = 822.8. CD4 effect: OR = 1.000 (95% CI: 1.000–1.001; *p* = 0.23). HEV, Hepatitis E Virus; PLWH, People Living with HIV; CD4, Cluster of Differentiation 4.

**Table 1 viruses-17-01038-t001:** Characteristics of hepatitis E virus infection in 670 PLWH in Shanghai, China, 2018–2024.

Characteristics	Overall, *n* = 670	Anti-HEV IgM/IgG		*p*-Value
		Negative, *n* = 468 (69.85%)	Positive, *n* = 202 (30.15%)	
Age (years)	40.00 [32.00, 53.00]	37.00 [31.00, 50.00]	47.00 [38.00, 57.00]	<0.001
Gender, *n* (%)				0.754
Female	82 (12.24)	59 (12.61)	23 (11.39)	
Male	588 (87.76)	409 (87.39)	179 (88.61)	
Marital Status, *n* (%)				0.076
Unmarried ^1^	182 (27.16)	137 (29.27)	45 (22.28)	
Married	488 (72.84)	331 (70.73)	157 (77.72)	
Ethnicity, *n* (%)				0.207
Han	663 (98.96)	465 (99.36)	198 (98.02)	
Others	7 (1.04)	3 (0.64)	4 (1.98)	
WHO Stage, *n* (%)				0.677
Stage I/II	411 (61.34)	290 (61.97)	121 (59.90)	
Stage III/IV	259 (38.66)	178 (38.03)	81 (40.10)	
CD4 Count (cells/μL)	211.94 [45.20, 473.71]	196.90 [39.02, 491.21]	232.43 [77.45, 443.52]	0.273
<200, *n* (%)	323 (48.21)	235 (20.21)	88 (43.56)	
≥200 and <400, *n* (%)	139 (20.75)	92 (19.66)	47 (23.27)	
≥400, *n* (%)	208 (31.04)	141 (30.13)	67 (33.17)	
ART, *n* (%)				0.322
N + N + NN ^2^	162 (24.18)	107 (22.86)	55 (27.23)	
N + N + PIs ^3^	33 (4.93)	20 (4.27)	13 (6.44)	
N + N + NR ^4^	308 (45.97)	219 (46.79)	89 (44.06)	
None	167 (24.93)	122 (26.07)	45 (22.28)	
Viral Hepatitis Co-infections				0.209
HBV co-infection, *n* (%)	46 (6.87)	27 (5.77)	19 (9.41)	
HCV co-infection, *n* (%)	27 (4.03)	18 (3.85)	9 (4.46)	
No virus, *n* (%)	597 (89.10)	422 (90.17)	175 (86.63)	
Chronic Liver Disease, *n* (%)				<0.001
Liver cirrhosis	19 (2.84)	4 (0.85) ^5^	15 (7.43) ^6^	
No liver cirrhosis	651 (97.16)	464 (99.15)	187 (92.57)	
Liver Function Tests				
ALB (g/L)	38.25 [31.80, 44.48]	38.16 [31.64, 44.70]	38.33 [32.00, 43.37]	0.587
AST (U/L)	30.00 [20.00, 63.75]	29.00 [20.00, 57.00]	31.00 [19.85, 77.53]	0.354
ALT (U/L)	30.00 [16.00, 79.38]	29.40 [16.00, 75.00]	34.00 [16.00, 94.30]	0.112
TBIL (μmol/L)	10.40 [7.01, 16.70]	9.85 [6.90, 15.10]	11.60 [7.43, 19.65]	0.003
Anti-HEV IgM (S/CO)	0.07 [0.03, 0.10]	0.06 [0.02, 0.10]	0.10 [0.04, 0.16]	<0.001
Anti-HEV IgG (S/CO)	0.13 [0.07, 1.75]	0.10 [0.04, 0.16]	3.65 [2.02, 7.08]	<0.001

^1^ Includes unmarried, divorced, and widowed; ^2^ includes two nucleoside reverse transcriptase inhibitors (NRTIs) and a non-nucleoside reverse transcriptase inhibitor (NNRTI); ^3^ includes two NRTIs and a protease inhibitor (PI); ^4^ includes two NRTIs and an integrase chain termination inhibitor (INSTIs); ^5^ 2 patients had HBV-related cirrhosis, 1 patient had HCV-related cirrhosis, and 1 patient had cirrhosis due to other causes; ^6^ 7 patients had HBV-related cirrhosis, 2 patients had HCV-related cirrhosis, and 6 patients had cirrhosis due to other causes. HEV, Hepatitis E virus; HIV, Human immunodeficiency virus; PLWH, People living with HIV; ART, Antiretroviral therapy; HBV, Hepatitis B virus; HCV, Hepatitis C virus; ALB, Albumin; AST, Aspartate aminotransferase; ALT, Alanine transaminase; TBIL, Total bilirubin; IgM, Immunoglobulin M; IgG, Immunoglobulin G; S/CO, Signal-to-cutoff ratio.

**Table 2 viruses-17-01038-t002:** HEV serology results.

Group	Anti-HEV Antibody	*n*	Percentage (%)
PLWH	Anti-HEV IgM-negative and lgG-negative	468	69.85
	Anti-HEV IgM-positive or IgG-positive	202	30.15
	Anti-HEV IgG-positive	201	30.00
	Anti-HEV IgG-positive and lgM-negative	194	28.96
	Anti-HEV IgM-positive	8	1.19
	Anti-HEV IgM-positive and IgG-positive	7	1.04
	Anti-HEV IgM-positive and IgG-negative	1	0.15
HIV-negative health-check attendees	Anti-HEV IgG-positive	82	17.67
	Anti-HEV IgM-positive	0	0

PLWH, People living with HIV; HEV, Hepatitis E virus; IgM, Immunoglobulin M; IgG, Immunoglobulin G.

**Table 3 viruses-17-01038-t003:** Demographic and clinical characteristics of patients with detectable HEV IgM.

No.	Date	Gender	Age	Anti-HEV IgM(S/CO)	Anti-HEV IgG(S/CO)	CD4 Count (Cells/uL)	ALB (g/L)	ALT (U/L)	AST (U/L)	TBIL (umol/L)
1	2018-05	Male	57	52.39	2.69	134.35	36.00	754.00	1294.00	141.60
2	2018-12	Male	30	39.06	4.14	106.18	36.00	553.00	753.00	156.40
3	2020-09	Male	52	28.00	5.47	319.99	32.72	673.00	63.00	100.30
4	2021-07	Male	31	25.43	3.56	156.15	38.35	1043.00	225.00	101.70
5	2021-12	Male	64	25.58	13.23	153.25	36.66	968.00	485.00	17.70
6	2022-08	Male	40	12.42	12.44	533.48	27.55	30.00	36.00	42.10
7	2024-07	Male	42	23.17	7.30	69.99	40.10	1618.90	1571.30	196.00
8	2024-09	Male	42	8.10	0.14	303.79	23.10	53.00	67.00	5.50

HEV, Hepatitis E Virus; HIV, Human Immunodeficiency Virus; ALB, Albumin; AST, Aspartate Aminotransferase; ALT, Alanine Transaminase; TBIL, Total Bilirubin; IgM, Immunoglobulin M; IgG, Immunoglobulin G; S/CO, Signal-to-Cutoff Ratio.

**Table 4 viruses-17-01038-t004:** Unifactorial and multifactorial analysis of HEV infections occurring in PLWH.

Characteristics	Univariate Analysis (*n* = 670)		Multivariate Analysis (*n* = 670)	
	Crude OR (95% CI)	*p*-Value	Adjusted OR (95% CI)	*p*-Value
Age	1.005 (0.993–1.017)	0.407	1.003 (0.990–1.016)	0.658
Gender				
Female	Base			
Male	0.982 (0.594–1.622)	0.943	0.876 (0.520–1.477)	0.620
Marital Status				
Married	Base			
Unmarried ^1^	0.869 (0.596–1.266)	0.464	0.832 (0.541–1.281)	0.404
Ethnicity				
Han	Base			
Others	0.000 (0.000–∞)	0.979	0.000 (0.000–∞)	0.978
WHO Stage				
Stage I/II	Base			
Stage III/IV	1.422 (1.016–1.988)	0.040	1.566 (1.068–2.297)	0.022
CD4 Count (cells/μL)	1.000 (0.999–1.000)	0.380	1.000 (0.999–1.000)	0.284
ART				
None	Base			
N + N + NN ^2^	0.987 (0.619–1.575)	0.957	0.913 (0.564–1.479)	0.712
N + N + PIs ^3^	0.913 (0.606–1.375)	0.663	0.893 (0.577–1.382)	0.612
N + N + NR ^4^	0.962 (0.427–2.165)	0.925	0.793 (0.339–1.857)	0.593
Viral Hepatitis Co-infections				
No virus	Base			
HBV co-infection	1.391 (0.746–2.593)	0.299	1.181 (0.600–2.322)	0.631
HCV co-infection	1.382 (0.621–3.073)	0.428	1.628 (0.674–3.928)	0.278
Chronic Liver Disease				
No liver cirrhosis	Base			
Liver cirrhosis	1.364 (0.529–3.517)	0.521	1.532 (0.575–4.082)	0.394
Liver Function Tests				
ALB (g/L)	1.002 (0.983–1.021)	0.851	1.020 (0.995–1.046)	0.123
AST (U/L)	1.000 (0.999–1.000)	0.928	1.000 (0.999–1.001)	0.614
ALT (U/L)	1.000 (1.000–1.000)	0.852	1.000 (0.999–1.001)	0.766
TBIL (μmol/L)	1.001 (0.999–1.004)	0.282	1.001 (0.998–1.004)	0.362

^1^ Includes unmarried, divorced, and widowed. ^2^ Includes two nucleoside reverse transcriptase inhibitors (NRTIs) and a non-nucleoside reverse transcriptase inhibitor (NNRTI). c Includes two NRTIs and a protease inhibitor (PI). ^3^ Includes two NRTIs and an integrase chain termination inhibitor (INSTI); ^4^ includes two NRTIs and an integrase chain termination inhibitor (INSTIs). OR, Odds ratio; 95% CI, 95% Confidence interval; HEV, Hepatitis E virus; HIV, Human immunodeficiency virus; PLWH, People living with HIV; ART, Antiretroviral therapy; HBV, Hepatitis B virus; HCV, Hepatitis C virus; ALB, Albumin; AST, Aspartate aminotransferase; ALT, Alanine transaminase; TBIL, Total bilirubin; IgM, Immunoglobulin M; IgG, Immunoglobulin G; S/CO, Signal-to-cutoff ratio.

## Data Availability

No new data were created.

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
