# Peer review of "Clinical Characteristics and Epidemiological Features of Hepatitis E Virus Infection Among People Living with HIV in Shanghai, China"

_viruses, 2025, doi:10.3390/v17081038_

Round 1
Reviewer 1 Report
Comments and Suggestions for Authors
The manuscript by Zhao et al. entitled Clinical Characteristics and Epidemiological Features of Hepatitis E Virus Infection among People Living with HIV in Shanghai, China,” narrows an important geographical and clinical knowledge gap on HIV/HEV co-infection in eastern China. Using a large single-center cohort of 670 people living with HIV (PLWH) and 464 HIV-negative controls, the authors demonstrate that anti-HEV IgG seroprevalence is significantly higher in PLWH (30.0 %) than in controls (17.7 %). Multivariable analysis shows that advanced WHO HIV stage (III/IV) independently predicts HEV seropositivity, whereas age, sex, and CD4 count do not. By directly estimating the excess HEV burden attributable to HIV infection, the study complements existing data, which are largely drawn from western and southwestern provinces in China, and reveals novel and meaningful east-coast heterogeneity. Clinically, the findings strongly support the need for routine HEV screening in PLWH who present with unexplained elevations in liver enzymes. Despite many strengths of the current studies, the manuscript in its present form has several conceptual, methodological, and presentational shortcomings that should be addressed to enhance the novelty and scientific clarity of the work. To allow for a comprehensive revision of the current manuscript, please find below a thematically organized flow that should improve the study.
Major Critiques:
-The manuscript employs a retrospective, cross-sectional design, yet several phrases imply causality. For example, the sentence advanced stage emerged as a risk factor that increases HEV susceptibility attributes a directional effect that cannot be inferred without longitudinal data. To maintain methodological consistency, all such wording should be reframed in associative terms and should acknowledge potential reverse causation (e.g., chronic HEV infection possibly hastening HIV progression).
Suggestions for revisions to the following text: Replace“advanced stage emerged as a risk factor that increases HEV susceptibility” with Advanced WHO HIV stage (III/IV) was associated with a higher likelihood of prior HEV exposure; however, the cross-sectional design precludes inference about directionality, and unrecognised chronic HEV infection could itself contribute to HIV disease progression.
-The manuscript’s description of population selection lacks critical details, making it difficult to judge the representativeness of both PLWH and control groups. Without knowing whether anti-HEV tests were ordered routinely or only for symptomatic individuals, readers cannot gauge possible selection bias. Likewise, the control group is ambiguously labelled “healthy” in the Abstract yet described as “hospital attendees” in the Methods, leaving their true exposure context unclear. These ambiguities weaken the study’s internal validity and may distort prevalence estimates. A succinct flow chart showing the sampling selection process, plus precise terminology for controls and additional covariate adjustment, would resolve these issues. The following suggestions can be made to address these concerns:
- Clarify why PLWH were tested
Current text (Methods, first paragraph):“Between 2018 and 2024, 670 adult PLWH underwent anti-HEV testing.”
Consider replacing with or revising the text as follows: “Between January 2018 and May 2024, our clinic followed 1,944 PLWH; of these, 670 (34.5 %) underwent anti-HEV testing as part of routine annual liver-function surveillance, independent of clinical hepatitis symptoms.”
- Add a step-by-step flow diagram that clarifies the rationale for sampling and testing of study subjects.
As a suggestion, immediately after the revised sentence above, I would suggest a Figure 1 that best captures the following: Figure 1 depicts the full sampling structure, including total PLWH in care, numbers tested, exclusions, and the final analytic cohort.
- The control groups should be better defined as the current manuscript stated, “Healthy controls (n = 464) were recruited from the same hospital.” It is unclear if the healthy controls, as mentioned, came to the hospital for other reasons or if these are routine hospital visits. As such please consider replacing with: Controls comprised 464 HIV-seronegative adults who attended the hospital’s annual employment health-check program and reported no chronic liver disease or immunodeficiency. Alternatively, there needs to be a justification of what clinical parameters would have prompted testing or are these routine tests?
- Please try to harmonize terminology throughout the manuscript by searching and replacing “healthy controls” with HIV-negative health-check attendees to maintain consistency.
- There is a need to also address potential confounding by potentially adding an additional sentence to description of the control subjects: As healthcare contact and transfusion history may influence HEV exposure, multivariable models were additionally adjusted for transfusion, occupation, and household income; variables with p < 0.10 in univariate analysis were retained.
Adding a flow diagram and indicating revisions above will make the sampling framework transparent and justify case versus control comparability. As these are suggestions, the authors can adopt their own approach to addressing these critiques.
-The clinical Diagnostic methods are appropriate but incompletely reported. While the diagnostic approach is broadly appropriate, key performance details relevant to the testing kits are missing. The manuscript names the Wantai ELISA kit but omits its published sensitivity and specificity, and it reports HEV-RNA testing only in IgM-positive sera with a 500 copies mL-¹ limit of detection (LOD). From a testing standpoint, this LOD threshold might not capture low-level viremia in immunocompromised patients. Therefore, there is the possibility of false negatives with the testing kit, giving the lower limit for detection might not capture low viremic cases. Including the characteristics of the detection kit and acknowledging the risk of false-negative PCR results would give readers a clearer picture of diagnostic certainty.
The authors should consider the following suggested fixes for the manuscript revision.
- Methods, especially the Serological testing.
Current text: Anti-HEV IgG and IgM were detected using a Wantai ELISA kit.
Consider replacing with: Anti-HEV IgG and IgM were measured using the Wantai ELISA kit (Beijing Wantai, China; reported sensitivity 99.6 % and specificity 98.9 %).
- Methods specifically on viral RNA testing
Current text: HEV RNA was assessed only in IgM-positive sera using a commercial RT-qPCR (LOD 500 copies mL-¹).
Consider replacing with: HEV RNA was tested solely in IgM-positive sera by commercial RT-qPCR, with a lower limit of detection of 500 copies mL-¹.
- Discussion: In the discussion, please consider adding a cautionary note after reporting negative PCR results because there are possibilities of false negatives if the viral load is below the threshold for detection by the testing approach used in this study.
Suggested inclusion in the discussion: Given this 500 copies mL-¹ threshold, low-level viremia may have gone undetected, so a negative RNA result does not definitively exclude chronic HEV infection in PLWH.
-Statistical analysis is generally sound, but some modelling choices need additional refinement. The manuscript states that all variables with p < 0.05 in univariate testing were forced into multivariate logistic regression; however, WHO stage and CD4 count are co-linear indicators of immune status. As such, variance-inflation factors or goodness-of-fit indices should be provided to demonstrate model stability. A more informative approach might stratify analyses by antiretroviral treatment status or duration of viral suppression(HIV+ but undatable viral load), because immune reconstitution could plausibly modulate HEV exposure or seroconversion probabilities. In addition, reporting crude and adjusted odds ratios side-by-side (rather than in separate tables) would assist readers in evaluating confounding.
-The Results section is very much populated with numeric estimates but occasionally lacks narrative context. For example, Figure 2 illustrates no significant seroprevalence trend from 2018-2024, yet testing frequency per year is not reported. Thus, if fewer tests were performed early on, the absence of a consistent trend in testing may be artefactual. Likewise, the authors describe a “non-monotonic pattern” across age groups but do not provide statistical post-hoc pairwise comparisons or regression analysis to support that assertion. Additionally, while the CD4-versus-risk curve in Figure 3 is visually informative, its legend should specify the statistical evaluation and confidence-interval computation.
-The Discussion is comprehensive but would benefit from better structuring. A succinct first paragraph recapping the novel findings from this research will orient readers before elaborating in greater detail on comparisons with other external cohorts. Furthermore, several statements within the text over-interpret the descriptive data. For example, suggesting that persistent intestinal mucosal damage “likely constitutes a biological risk factor” goes beyond the evidence presented. Please consider rephrasing as “may constitute,” and citing published mechanistic literature on microbial translocation in late-stage HIV would keep the discussion balanced. When benchmarking prevalence against other Chinese provinces, it is important to acknowledge potential variability and differences in the diagnostic sensitivity and specificity of the testing assay used. The current text comes across as being equivalent despite the possibility of methodological diversity in the testing kits used.
-The authors did acknowledge Limitations to their current study, but an important one is omitted. The complete reliance on serology alone risks underestimating HEV in PLWH because impaired humoral responses can delay or abrogate antibody appearance. The authors should discuss the extent of potential misclassification and how bias results (likely towards the null) can occur. The authors might also consider future incorporation of HEV-specific T-cell assays or antigen detection to capture seronegative infection.
-Figures, tables, and supplementary material are generally clear. Nevertheless, Table 4 combines univariate and multivariate outputs but omits the number of observations used in each model. As such, readers cannot verify whether list-wise deletion reduced the sample size. Figure 1’s flow chart should display absolute numbers at each exclusion step, enhancing clarity and reproducibility. The supplementary Table S1 is valuable; referencing it explicitly when discussing global prevalence would help readers locate the comparative data.
Minor comments:
-The language and style are mostly fluent, yet several minor grammatical lapses merit attention (e.g., No one in the healthy control group tested positive….. rather than - no one in the healthy control group was positive). Occasional verb-tense inconsistencies (“data were downloaded” vs “clinical data is”) should be standardized. Some acronyms are listed in an abbreviations section, which is appreciated, but the first use of “PLWH” lacks definition in the Abstract.
-The References are meticulously provided, with >70 citations integrated. Still, very recent Chinese epidemiological reports (e.g., Lu et al., 2025, Infect Dis) are discussed in the Results yet are absent from the numbered list. Please cross-check the numbering to prevent discordance.
While this research manuscript centers on a very important medical problem and is well written overall, addressing critiques/suggestions detailed above will substantially enhance the methodological robustness, interpretability and international relevance of the study, positioning it as a credible contribution to the growing literature on HEV in immunocompromised populations.
Comments on the Quality of English Language
Aspects on how to improve the English within certain parts of the text are included in my review comments.
Reviewer 2 Report
Comments and Suggestions for Authors
Dear Authors,
Thanks for sending me this paper for evaluation. In general, the paper shows merit and gives new information about HEV seroprevalence in people living with HIV. I would like to address some issues to improve the quality;
Methods:
- There is no information about sensitivity and specificity of enzyme immunoassay that could impact in the result.
- There is no information about HEV RNA testing. Is it in house? What is the reference? Limit of detection and sensitivity?
Results:
- In general, the results are well presented. However, the information about prevalence of anti-HEV in control group could be included in tables.
Discussion:
In this section, authors gave more information about the study. However some references about HEV prevalence in other settings could be included to compare the results founded.
For example, this prevalence was much higher than found in other regions of the world. Please see and include the following references?
- https://pubmed.ncbi.nlm.nih.gov/36603828/
- https://pubmed.ncbi.nlm.nih.gov/39602852/
- https://pubmed.ncbi.nlm.nih.gov/35476107/
- https://pubmed.ncbi.nlm.nih.gov/39961224/
- https://pubmed.ncbi.nlm.nih.gov/39057763/
Reviewer 3 Report
Comments and Suggestions for Authors
In " Clinical Characteristics and Epidemiological Features of Hepatitis E Virus Infection Among People Living with HIV in Shanghai, China ", Conglin Zhao et al investigated HEV sero prevalence, clinical characteristics, and associated risk factors in people living with HIV (PLWH) in Shanghai, China. The frequency of acute HEV infection among PLWH in Shanghai is low, the rate of past HEV exposure is substantial. The manuscript has plenty of data and interest,
several issues are need to be addressed.
- Table 2.HEV serology results. IgM antibody has existed for a relatively short period of time. Has the author considered nucleic acid testing.
- The author did not take into account the HEV infection situation in the population during the discussion. In fact, there is a lack of support for hev infection in the population. As a result, the response is discounted, and it is suggested to increase the infection status of people with weakened immunity.
- HEV infections mostly originate from animals. Did the author consider whether they had contact with animals, rural areas, or people engaged in animal business
- PLWH have a weakened immune system. Compared with other diseases that are obviously secondary to HIV, such as Pneumocystis carinii, HEV should be a secondary infection.
